# Connectivity-based Cerebrovascular Segmentation in Time-of-Flight Magnetic Resonance Angiography

## ABSTRACT

Accurate segmentation of cerebrovascular structures from TOF-MRA is vital for treating cerebrovascular diseases. However, existing methods rely on voxel categorization, leading to discontinuities in fine vessel locations. We propose a connectivity-based cerebrovascular segmentation method that considers inter-voxel relationships to overcome this limitation. By modeling connectivity, we transform voxel classification into predicting inter-voxel connectivity. Given cerebrovascular structures' sparse and widely distributed nature, we employ sparse 3D Bi-level routing attention to reduce computational overhead while effectively capturing cerebrovascular features. To enhance directional information extraction, we utilize the 3D-direction excitation block. Additionally, the 3D-direction interactive block continuously augments direction information in the feature map and sends it to the skip connection. We compare our method with current state-of-the-art cerebrovascular segmentation techniques and classical medical image segmentation methods using clinical and open cerebrovascular datasets. Our method demonstrates superior performance, outperforming existing approaches. Ablation experiments further validate the effectiveness of our proposed method.

## CCS CONCEPTS

• **Applied computing → Health informatics**.

## KEYWORDS

cerebrovascular segmentation, connectivity, directional information

## 1 INTRODUCTION

High morbidity and mortality rates associated with cerebrovascular diseases pose significant threats to health and well-being. Early and accurate diagnosis, coupled with effective treatment, is paramount in mitigating the progression of these conditions.

Current medical imaging technologies, including vascular ultrasound, computed tomography angiography, digital subtraction angiography, and magnetic resonance angiography (MRA), play a crucial role in early cerebrovascular disease diagnosis. Time-of-flight MRA (TOF-MRA) is preferred due to its non-radioactive, non-invasive nature and other advantages [11]. However, precise segmentation of cerebrovascular structures from TOF-MRA remains

*ACM MM, 2024, Melbourne, Australia*
© 2024 Copyright held by the owner/author(s). Publication rights licensed to ACM.
ACM ISBN 978-x-xxxx-xxxx-x/YY/MM
https://doi.org/10.1145/nnnnnnn.nnnnnnn

challenging, with manual segmentation being time-consuming and reliant on expert knowledge.

Deep learning has become increasingly prevalent in cerebrovascular segmentation due to its robust feature extraction capabilities[2, 6, 14]. However, many current deep learning-based approaches[9, 15, 16, 21] treat cerebrovascular segmentation as a voxel-level classification task, assigning each voxel a score indicating its likelihood of belonging to the cerebrovascular structure. While these methods have demonstrated favorable segmentation results, they often overlook the interrelation between voxels, leading to poor connectivity in delicate cerebrovascular structures. This limitation could potentially impact diagnostic accuracy and treatment planning.

Recent research has explored novel approaches to address these limitations, such as modeling inter-voxel relationships using connectivity masks [19]. Unlike traditional segmentation masks, connectivity masks incorporate directional information and guide the network to extract richer contextual information. This approach shows promise in enhancing cerebrovascular segmentation accuracy and improving diagnostic and treatment outcomes.

Each channel in the connectivity mask corresponds to a specific direction[26], enriching it with abundant directional information. Similarly, the feature map within the connectivity-based network also contains rich directional information. Extracting and leveraging this latent directional information from the feature map holds promise for enhancing the network's segmentation performance. To accomplish this, we introduce a 3D direction excitation block designed to amplify directional information from the feature map through channel-wise operations.

Cerebrovascular, which exhibits a widespread distribution across the brain[22]. Given the transformer model's capability to capture long-distance dependencies[8], employing it to extract cerebrovascular features is advantageous. However, cerebrovascular are sparsely distributed[10], leading to excessive computational overhead in background areas when using traditional transformers. We adopt a sparse attention mechanism,3D Bi-level routing attention, to address these concerns and devise a 3D Bi-level Former block. This approach prioritizes feature extraction in regions with dense blood vessel concentration, thereby mitigating computational burden while preserving the ability to capture long-range dependencies of cerebrovascular.

In summary, we propose a connectivity-based cerebrovascular segmentation framework, TransConNet, that addresses the main challenges of cerebrovascular segmentation by incorporating a connectivity mask. Our main contributions are as follows:

- Considering the propensity for voxel-based classification methods to cause disconnects in the thin cerebrovascular, we transformed the task of cerebrovascular segmentation from voxel classification to predicting inter-voxel connectivity.
- Considering the beneficial hidden directional information within feature maps for connectivity prediction, we devised

a 3D direction excitation block and a 3D direction interactive block to extract and facilitate the flow of hierarchical directional information between the encoder and decoder.

- Our experimental results demonstrate superiority on clinical and public datasets compared with four state-of-the-art cerebrovascular segmentation models and two classic medical segmentation methods.

## 2 RELATE WORK

This section briefly reviews the state-of-the-art approaches for cerebrovascular segmentation and the connect-based segmentation method.

### 2.1 Cerebrovascular Segmentation

Deep learning has rapidly emerged as the preferred method for cerebrovascular segmentation due to its exceptional ability to represent cerebrovascular features accurately [23]. Yet, some challenges in cerebrovascular segmentation remain, such as class imbalance.

The foreground region of the vessels is often significantly smaller than the background, leading to a class imbalance issue. Kervadec et al. [10] proposed boundary loss, which utilizes integrals over the interface between regions to complement regional information by penalizing misclassifications at boundary regions. Additionally, Wang et al. [24] introduced a novel method by embedding image composition generated by maximum intensity projection into 3D MRA volumetric images. This technique enhances the model's ability to capture subtle vascular features and improve segmentation accuracy. While these methods mitigate class imbalance issues, they still focus on computations in the background area. To optimize efficiency, we propose leveraging sparse attention to prioritize feature extraction in the blood vessel region, thus minimizing computational waste.

### 2.2 connect-based segmentation

Preserving the topology of cerebrovascular structures, encompassing their spatial arrangement and connectivity, is crucial for accurately segmenting cerebrovascular systems with complex and intricate features. Banerjee et al.[1] proposed a multi-task deep convolutional neural network that simultaneously learns the distance transform for voxels on the vessel tree surface and the vessel centerline, these auxiliary tasks contribute to preserving the topography of vessel segmentation. Similarly, Gupta et al. [7] introduced a topological interaction module to identify critical pixels that induce topological errors, further enhancing segmentation accuracy. However, despite the advancements made by these methods in increasing precision and accuracy, they still treat each voxel as an independent entity and do not explicitly consider inter-voxel relationships, which are crucial for fully preserving cerebrovascular connectivity.

A promising approach is connectivity-based segmentation methods, which leverage connectivity masks as training labels. These masks typically consist of 4, 8, or 26 channels, depending on each pixel's predicted number of neighboring pixels, encoding information about voxel connectivity in designated directions. Kampffmeyer et al. cite kampffmeyer2018connnet pioneered four and 8-channel connection masks, demonstrating their efficacy in natural image

segmentation. Subsequently, Qin et al.[19] explored connectivity models, showcasing their effectiveness in 3D medical image segmentation. Furthermore, Yang et al.[27] integrated pixel connectivity prediction and pixel classification for in vivo human esophageal optical coherence tomography layer segmentation.

## 3 METHDOS

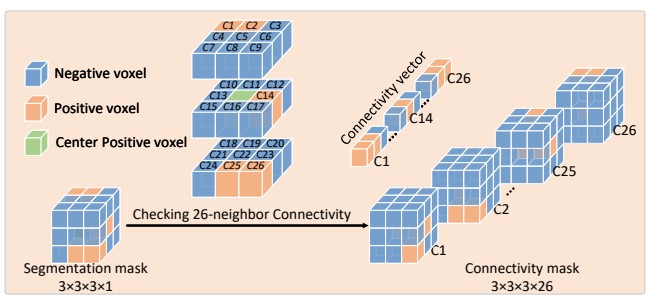

**Figure 1: Connectivity modeling process**

### 3.1 connectivity modeling

In a three-dimensional MRA, 26-connectivity effectively describes the relationship between one voxel and its 26 neighbors (see Fig. 1). Given a voxel at position $p = (x, y, z)$, we denote its 26 neighbor voxels as $q_i = (x + u, y + v, z + w)$, $i \in c_1, c_2, \cdots, c_{26}$, where $u, v, w \in 0, \pm 1$ and not all are 0. If both $p$ and $q_i$ are vessel voxels, their pair $(p, q_{c_i})$ is considered connected, and the corresponding $c_i$th channel of position $p$ is marked positive. Otherwise, it is marked negative for a disconnected pair $(p, q_{c_i})$. By traversing each voxel of the segmentation mask, we establish the relationship between it and the adjacent 26 voxels, resulting in a 26-channel connectivity mask. Zero padding is performed on the borders of the MRA volume to maintain the size of the generated connectivity mask. By representing the segmentation mask as a connectivity mask, we transform the task of voxel classification into inter-voxel relationship prediction. We utilize both the segmentation mask and connectivity mask to model voxel correlation for supervised learning.

### 3.2 Network Architecture

The backbone of the proposed network is presented in Fig. 2. The proposed TransConNet is an asymmetrical encoder-decoder structure, mainly composed of encoder, bottleneck, decoder and skip connections, which can learn the hierarchy of connectivity features at multiple scales.

In the encoder, the patch partition block initially divides the input images into non-overlapping patches of size $4 \times 4 \times 4$. Subsequently, a linear patch embedding block is applied to expand the feature dimension to $3C$. Here, $C = 26$, representing the dimension of the 26-adjacent connection. Each scale of the encoder is composed of several Bi-former blocks and patch merging blocks, of which patch merging block downsamples spacial resolution and increasing channel dimensions, while Bi-former blocks focus on connectivity feature learning. Ultimately, the encoder transforms features to the

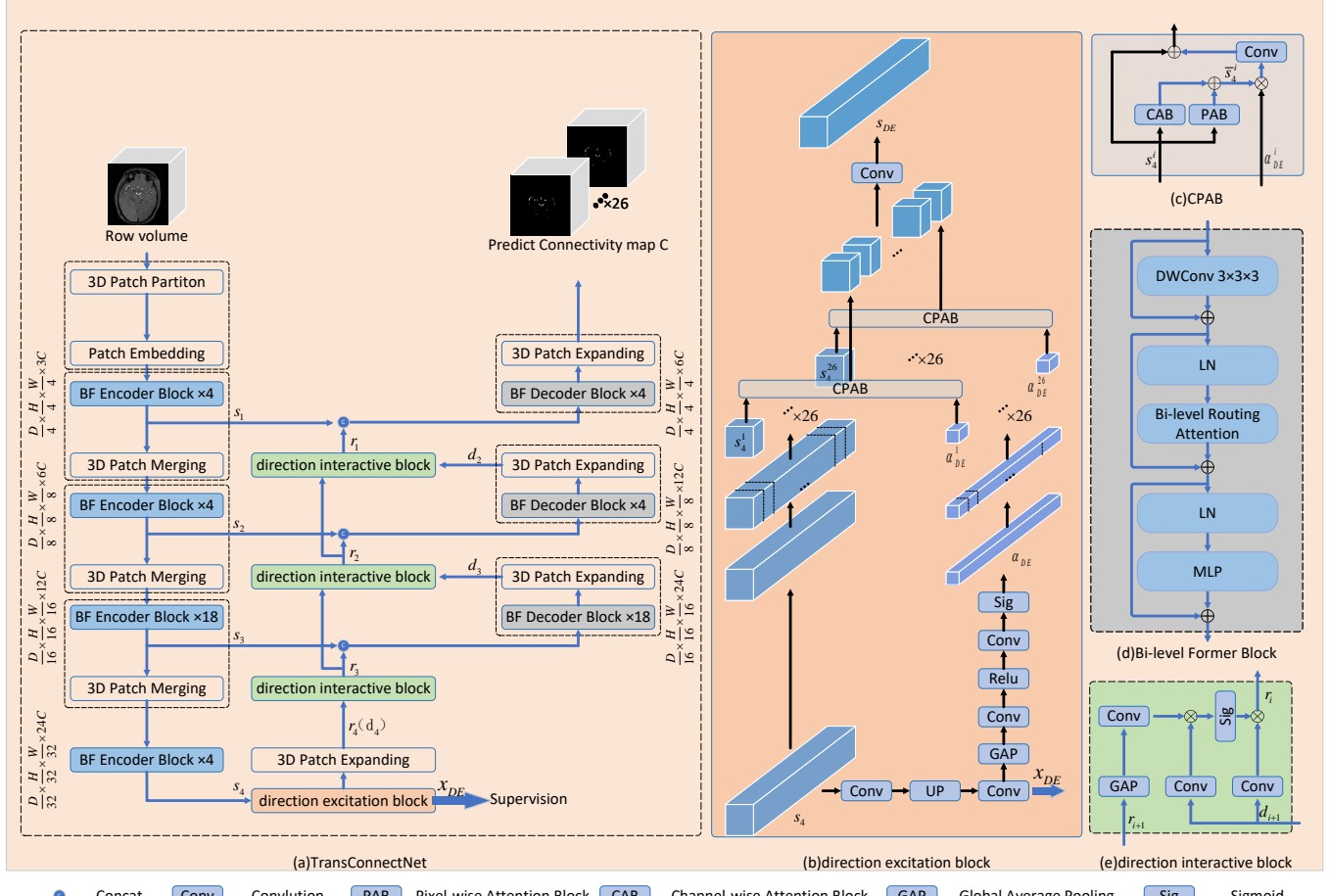

Figure 2: Architecture of TransConNet for Cerebrovascular Segmentation.

size of $\frac{W}{32} \times \frac{H}{32} \times \frac{S}{32} \times 24C$. In contrast to patch merging block, the decoder utilizes patch expanding block to decompress the feature until spacial resoultion matches the input image. The final patch expanding block restore the feature maps to the size of $W \times H \times S \times C$.

To capture directional information stored in the channel dimension, 3D direction excitation is applied at the bottleneck. Additionally, each scale incorporates a 3D-direction enhancement block in the skip connection to facilitate the flow of hierarchical direction information between the encoder and decoder. Further elaboration on TransConNet's details follows.

*3.2.1 3D Bi-level Former block.* The Transformer model demonstrates its remarkable ability to capture long-range dependencies, making it well-suited to model complex vascular structures that continuously connect different brain regions. However, the cerebrovascular target is sparse and small compared to the background tissue. Thus, we utilize 3D Bi-level routing attention in our 3D BiFormer block to improve the computation efficiencies. It utilizes a sparse attention mechanism to compute the self-attention feature, which can effectively process the sparse targets within the denser background.

Specifically, the Bi-level routing attention is done as follows. Given an input feature map $X \in \mathbb{R}^{H' \times W' \times D' \times C'}$, we first divide it into $S \times S \times S$ nonoverlapped patches, and reshape it as $X_r \in \mathbb{R}^{S^3 \times \frac{H'W'D'}{S^3} \times C'}$. Then, we derive the query, key,and value tensor, $Q, K, V \in \mathbb{R}^{S^3 \times \frac{H'W'D'}{S^3} \times C'}$, with linear projetions:

$$(Q, K, V) = (X_r W_Q, X_r W_K, X_r W_V), \quad (1)$$

where $W_Q, W_K, W_V \in \mathbb{R}^{C' \times C'}$ are projection weights for the query, key, and value, respectively.

After that, we seek the dynamic sparsity through the construction of a directed graph. Specifically, the region-level queries and keys, $Q_r, K_r \in \mathbb{R}^{S^3 \times C'}$, are derived by applying per-region average on $Q$ and $K$ respectively. We then calculate the adjacency matrix $M_r \in \mathbb{R}^{S^3 \times S^3}$ as follow:

$$M_r = Q_r (K_r)^\top, \quad (2)$$

which gives the correlation between different regions. Then, we use row-wise top-k operator to retain only the first $k$ connections of each region (k defaults to 4) and obtain the index matrix $I_r \in \mathbb{R}^{S^3 \times k}$:

$$I_r = \text{TopK}(M_r). \quad (3)$$

We then extract the corresponding key and value tensor of interest through the gather operator:

$$K_g = \text{Gather}(K, I_r), V_g = \text{Gather}(V, I_r), \tag{4}$$

where $K_g, V_g \in \mathbb{R}^{S^3 \times \frac{kH'W'D'}{S^3} \times C'}$ are gathered key and values tensor. Further, we apply attention on gathered key-value paris as:

$$O = \text{Softmax}(\frac{QK_g^{\top}}{\sqrt{C'}})V_g. \tag{5}$$

Finally, we reshape the output attention feature $O \in \mathbb{R}^{S^3 \times \frac{kH'W'D'}{S^3} \times C}$ back to the size of $H' \times W' \times D' \times C'$.

The designed 3D Bi-level Former block is like the standard vision transform structure, which is composed of depthwise convolution (DWConv), multi-layer percetron (MLP), and Bi-level routing attention (BiRA).

Supposing the input feature as $P_{\text{in}}$, the previous calculaiton process can be defined as follows:

$$P_1 = \text{DWConv}(P_{\text{in}}) + P_{\text{in}}, \tag{6}$$

$$P_2 = \text{BiRA} \circ \text{LN}(P_1) + P_1, \tag{7}$$

$$P_{\text{out}} = \text{MLP} \circ \text{LN}(P_2) + P_2. \tag{8}$$

First, the embedded token sequence of the input images undergoes layer normalization to ensure that the input values are within manageable ranges. Then, it is passed through the PDSA module, with the residual connecting its output to the embedded sequence. Following this, another layer normalization is applied, and the feedforward network employs an MLP for feature projection, followed by a residual connection.

The structure of the designed 3D BiFormer block is shown in Figure.2(d). consisting of three main components: depthwise convolution (DWConv), multi-layer percetron (MLP), and Bi-level routing attention (BiRA).

### 3.2.2 3D-direction excitation block.
We incorporate the connectivity mask during the training process, to capture the directional information within the channels of the feature map. Building upon this approach, we introduce the 3D direction excitation (DE) Module, where directional information is predominantly captured through channel-level operations. The DE block is depicted in Figure.2(b).

Suppose $s_i$ as the output of the $i$-th scale of the encoder. Then $s_4$ is sent to the bottleneck, i.e. direction excitation block. Two branches are included, the first branch is for weights for each direction channel, and the second branch is the corresponding direction feature.

For the first branch, we first upsample $s_4$ to the input size and get a preliminary output $x_{DE}$, which will be supervised to learn the connectivity mask. Then, we use global average pooling to squeeze $x_{DE}$ into a vector $\alpha_{DE}$ of size $1 \times 1 \times 1 \times 24C$.

$$x_{DE} = (\text{Conv} \circ \text{Upsample} \circ \text{Conv})(s_4), \tag{9}$$

$$\alpha_{DE} = (\text{Sig} \circ \text{Conv} \circ \text{Conv} \circ \text{Avgp})(x_{DE}). \tag{10}$$

Then, we split the latent features $s_4$ into 26 parts by channel wise slicing, of which the $i$th slices is denoted as $s_4^i$. We pass each feature slice $s_4^i$ through pixel-wise and channel-wise attention blocks to capture the long-range and inter-channel dependencies, results in $\bar{s}_4^i$:

$$\bar{s}_4^i = \text{PAB}(s_4^i) + \text{CAB}(s_4^i). \tag{11}$$

Then, we stack all $\bar{s}_4^i$ by channel-wise dimension into one feature map $\bar{s}_4$. We then multiply $\bar{s}_4$ and $\alpha_{DE}$ in a channel-wise manner, allowing network to learn specific direction information.

The final output of 3D-direction excitation block is recoded as:

$$s_{DE} = \text{Conv}(\text{Conv}(\bar{s}_4 \cdot \alpha_{DE}) + s_4). \tag{12}$$

### 3.2.3 3D-direction interactive block.
To make the direction information fused into the feature map in each scale of the decoder, we incorporate a direction interactive block (DIB) to transmit direction information from the down-scale to the up-scale of the decoder. The direction interactive block as show in Figure.2(e)

At the $i$th scale, DIB fuse the main feaure map $d_{i+1}$ that from decoder and direction enhanced map $r_{i+1}$ .Suppose the output of the $i$th DIB as $r_i$,in order to avoid the interference of spatial information, we first apply Global Average Pooling on $r_{i+1}$, and project it to the same number of channels as $d_{i+1}$:

$$\bar{r}_{i+1} = (\text{Conv} \circ \text{Avgp})(r_{i+1}). \tag{13}$$

Next, we dot $\bar{r}_{i+1}$ with the encoded $d_{i+1}$ channel-wise, resulting in $\bar{d}_{i+1}$, where direction-related information is enhanced across channels. We then apply the sigmoid function to $\bar{d}_{i+1}$ and multiply it with the encoded $d_{i+1}$ to enhance the directional information on the main feature map $d_{i+1}$:

$$r_i = \text{Sig}(\bar{d}_{i+1}) \otimes \text{Conv}(d_{i+1}). \tag{14}$$

The direction-enhanced map $r_i$ is concatenated with $s_i$, the feature from the encoder layer, to enrich the directional information in the decoder feature map. Additionally, $r_i$ is passed to the subsequent DIB as input, facilitating the bottom-up flow of directional information.

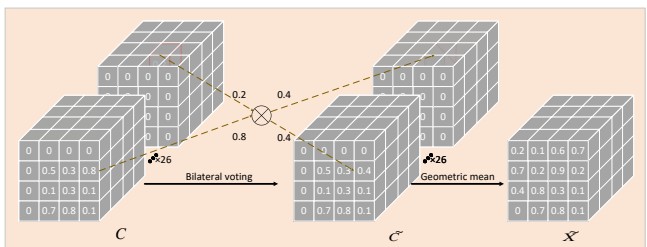

**Figure 3: The overview of Bilateral voting for connectivity output**

### 3.2.4 Bilateral voting for connectivity output.
In the predicted segmentation mask $S$, if a pair of neighboring voxels $S(x, y, z)$ and $S(x + a, y + b, z + c)$ belongs to vessel, where $a, b, c \in \{0, \pm1\}$ and are not all simultaneously 0, the connected pairs $C_i(x, y, z)$ and $C_{27-1}(x + a, y + b, z + c)$ in the predicted connectivity map $C$ should ideally be equal. This implies that the probabilities of unidirectional connection to each other should be equal, but in practice, this is rarely the case. Therefore, we propose 3D Bilateral Voting (BV)to address this issue.The process of bilateral voting is shown in the Fig. 3.

We multiply all connectivity pairs separately and reassign the resulting result to the connection pair, then we can obtain the new connectivity map $\tilde{C}$, and the process can be formulated as:

$$\tilde{C}_i(x, y, z) = \tilde{C}_{27-i}(x + a, y + b, z + c)$$
$$= \sqrt{C_i(x, y, z) \cdot C_{27-i}(x + a, y + b, z + c)}. \quad (15)$$

Here, $a, b, c \in \{0, \pm 1\}$ and are not all simultaneously 0. We take the segmentation prediction $\tilde{X}$ as the geometric mean of the first $b$ largest values of $\tilde{C}_{(i)}$, where $\tilde{C}_{(i)}$ denotes as the $i$th largest value of $\tilde{C}$. This can be expressed as:

$$\tilde{X} = (\prod_{i=1}^{b} \tilde{C}_{(i)})^{\frac{1}{b}}. \quad (16)$$

To obtain the final vessel segmentation mask $S$, we apply a thresholding operation to $\tilde{X}$ as follows:

$$S(x, y, z) = \begin{cases} 1 & \text{if } \tilde{X}(x, y, z) > 0.5, \\ 0 & \text{if } \tilde{X}(x, y, z) \le 0.5. \end{cases} \quad (17)$$

## 3.3 Training Loss

Both TransConNet and its bottleneck block (directional excitation block) output the connectivity and segmentation masks. Thus, our training loss consists mainly of two parts: main loss $\mathcal{L}_{main}$ and auxiliary loss $\mathcal{L}_{aux}$, corresponding to the final and bottleneck outputs of TransConNet, represented as:

$$\mathcal{L}_{\text{total}} = \mathcal{L}_{\text{main}} + 0.3 \mathcal{L}_{\text{aux}}. \quad (18)$$

In particular, $\mathcal{L}_{\text{main}}$ and $\mathcal{L}_{\text{aux}}$ have the same form, consisting of connectivity loss and segmentation loss. As an example, the main loss has the form:

$$\mathcal{L}_{\text{main}} = \mathcal{L}_{\text{seg}} + \mathcal{L}_{\text{con}}. \quad (19)$$

Moreover, $\mathcal{L}_{\text{seg}}$ is composed of the binary crossing entropy loss $\mathcal{L}_{\text{BCE}}$, the dice loss $\mathcal{L}_{\text{Dice}}$, and the intersection over union loss $\mathcal{L}_{\text{IoU}}$[13, 20], represented as:

$$\mathcal{L}_{\text{seg}} = \mathcal{L}_{\text{BCE}}(\tilde{X}, G_S) + \mathcal{L}_{\text{Dice}}(S, G_S) + \mathcal{L}_{\text{IoU}}(S, G_S), \quad (20)$$

where $G_s$ and $G_c$ are the ground truth segmentation label and its corresponding connectivity labels. $\tilde{X}$ and $S$ are computed from Eq. 16 and Eq. 17 respectively.

To ensure that our predicted connectivity masks remain consistent with the ground truth $G_C$, we utilize both $C$ and $\tilde{C}$ (Eq. 15) to compute the connectivity loss:

$$\mathcal{L}_{\text{con}} = \mathcal{L}_{\text{BCE}}(C, G_C) + \mathcal{L}_{\text{BCE}}(\tilde{C}, G_C). \quad (21)$$

By integrating these two connectivity loss functions, our model is trained to generate accurate connectivity masks. This leads to improved performance in cerebrovascular segmentation tasks.

# 4 EXPERIMENTS

## 4.1 Cerebrovascular datasets

*4.1.1 FAH-WMU dataset.* The FAH-WMU dataset comprises 82 cases of TOF-MRA volumes, acquired using a 3T Philips Medical MRI System at the First Affiliated Hospital of Wenzhou Medical University. The acquisition parameters include an echo time of 3.46

ms, a resolution of $560 \times 560 \times 140$, a repetition time of 23 ms, a flip angle of 18 degrees, and a voxel size of $0.38 \times 0.38 \times 0.70$, mm$^3$.

Two graduate medical students initially manually annotated the cerebral vessels in each TOF-MRA volume. Subsequently, three experienced cerebrovascular clinicians compared the initially annotated vessels with the original TOF-MRA volumes to correct any missing or incorrect vessels. Finally, a chief physician further examined the annotation results.

Before conducting the experiments, written consent was obtained from all participants, and ethical approval was obtained from the respective ethics committees at Zhejiang University of Technology and the First Affiliated Hospital of Wenzhou Medical University.

*4.1.2 IXI dataset.* The IXI dataset[3] is a public cerebrovascular dataset that comprises 45 TOF MRA volumes. These volumes were acquired using a 3T MRI scanner and followed standardized protocols. All volumes have dimensions of $1024 \times 1024 \times 92$ and a voxel size of $0.264 \times 0.264 \times 0.8mm^3$. The ground truth for each volume has been annotated voxel-wise under the supervision of multiple radiologists, each possessing over three years of clinical experience.

## 4.2 Experimental details

We randomly selected 62 cases from the FAH-WMU dataset and 25 from the IXI dataset for training. The remaining cases from both datasets were used for validation. Independent experiments were conducted on each dataset.

During the training phase, we applied random axis mirror flips with a probability of 0.5 for all three axes and random rotations ranging from 0 to 90 degrees with a probability of 0.5 to perform data augmentation. All MRA volume data were normalized with a zero mean and unit standard deviation.

Due to the high GPU memory requirement for processing the entire volume during training, we divided the segmentation mask $G_S$, connectivity mask $G_C$, and raw volume into sub-volumes with dimensions of $128 \times 128 \times 128$, which was then fed into our network.

During testing, we post-processed the predicted results of the sub-volume to obtain the complete cerebrovascular segmentation results.

We trained and tested the models using the PyTorch framework, utilizing the Adam optimizer with a fixed learning rate of $10^{-4}$. The experiments were performed on an Nvidia V100 GPU, leveraging its computational power for efficient model training and evaluation.

## 4.3 Evaluation metrics

We assessed model's segmentation performance using various metrics, including Dice Similarity Coefficient (DSC) [4, 29], 95th percentile Hausdorff Distance (HD95) [12], Average Symmetric Surface Distance (ASSD) [31], Relative Volume Error (RVE), and Sensitivity (SENS) [28]. These metrics collectively evaluate spatial overlap and geometric accuracy, providing insights into segmentation agreement, accuracy across the region, geometric precision, identification of positive cases, and volume difference. We obtained a comprehensive evaluation of segmentation performance by considering these metrics together.

## 4.4 Comparison with State-of-the-Art Methods

We compared TransConNet with four state-of-the-art deep-based cerebrovascular segmentation models (CSNet [15], RENet [30], ER-Net [25], and DSCNet [18]), as well as two classical medical segmentation methods (3DUnet [5] and VTUet [17]), on the FAH-WMU and IXI datasets. All models utilize the same experimental setup and data augmentation.

Table 1 and Table 2 illustrate the quantitative metrics for different models on the FAH-WMU and IXI datasets, respectively, with values representing averages on the testing dataset. For the FAH-WMU dataset, TransConNet achieves a DSC of 92.268%, ASSD of 0.695mm, SENS of 91.331%, and REV of 2.581%, outperforming the state-of-the-art methods in all measurements. Similarly, on the IXI dataset, TransConNet achieves a DSC of 83.197%, ASSD of 1.376mm, SENS of 82.323%, and REV of 7.144%, again surpassing the performance of other methods.

Cerebral blood vessels are highly intricate structures consisting of numerous small vessels, which poses a significant challenge for accurate extraction. As depicted in Fig. 4, like 3DUNet [5], VTUNet [17], and ERNet [25] struggle to identify delicate cerebrovascular structures accurately, while others such as CSNet [15], RENet [30], and DSCNet [18] exhibit interruptions along the middle positions. However, TransConNet demonstrates promising segmentation results, by accurately capturing small vessels and ensuring connectivity in delicate structures. This success is attributed to TransConNet's utilization of voxel connectivity.

**Table 1: Cerebrovascular segmentation performance of different models over FAH-WMU dataset, with the best results highlighted**

| Method | DSC | HD95 | ASSD | REV | SENS |
|---|---|---|---|---|---|
| 3DUNet | 88.077 | 7.051 | 1.367 | 13.79 | 82.298 |
| VTUNet | 88.767 | 5.694 | 1.136 | 12.62 | 83.802 |
| CS2Net | 89.263 | 3.215 | 0.879 | 16.321 | 82.043 |
| RENet | 90.388 | 2.068 | 0.820 | 8.805 | 82.007 |
| ERNet | 90.721 | 2.902 | 0.929 | 5.172 | 89.191 |
| DSCNet | 91.143 | 1.845 | 0.777 | 10.937 | 86.620 |
| TansConNet | **92.413** | **1.188** | **0.669** | **2.581** | **91.331** |

**Table 2: Cerebrovascular segmentation performance of different models over the IXI dataset, with the best results highlighted**

| Method | DSC | HD95 | ASSD | REV | SENS |
|---|---|---|---|---|---|
| 3DUNet | 77.916 | 17.486 | 2.435 | 21.116 | 69.864 |
| VTUNet | 76.040 | 18.882 | 2.63 | 30.956 | 64.365 |
| CS2Net | 72.810 | 19.861 | 2.786 | 31.093 | 61.658 |
| RENet | 76.295 | 17.366 | 2.455 | 28.117 | 65.675 |
| ERNet | 81.539 | 13.041 | 1.961 | 15.844 | 75.321 |
| DSCNet | 82.081 | 10.578 | 1.825 | 7.159 | 81.110 |
| TansConNet | **83.197** | **6.819** | **1.376** | **7.144** | **82.323** |

## 4.5 Ablation study

In this work, we propose a connectivity-based cerebral blood vessel segmentation method. The goal of our work is to use connectivity masks to improve the segmentation accuracy of cerebral blood vessels and maintain the connections of small blood vessels. In particular, we utilize connectivity mask and connectivity loss to maintain vessel segmentation consistency and introduce direction excitation block and direction interactive block to excite and flow directional information in feature maps. All of these operations are specially designed to improve the connectivity of vessel segmentation and to preserve detailed features as much as possible. We conducted three ablation studies to investigate the influence of connectivity masks and their associated connectivity losses on segmentation. Additionally, we analyzed the individual impact of each module on segmentation outcomes, as well as examined the effects of different loss functions employed during training on segmentation results.

**Table 3: Cerebrovascular segmentation performance of TansConNet when adding connectivity mask and connectivity loss,√ indicates the presence of this mask and corresponding loss, with the best results highlighted**

| seg mask | con mask | DSC | HD95 | ASSD | REV | SENS |
|---|---|---|---|---|---|---|
| √ | | 90.978 | 1.399 | **0.761** | 5.996 | 88.262 |
| √ | √ | **91.930** | **1.284** | 0.786 | **3.309** | **90.369** |

*4.5.1 Ablations study on connectivity mask and connectivity loss.* In this experiment, we initially utilize the segmentation mask as the training label, followed by the addition of the connectivity mask and corresponding connectivity loss. As shown in Table 3, the addition of the connectivity mask and the corresponding connectivity loss increased the DSC by 0.952. Although there was a slight rise in ASSD, other indicators demonstrated improvement. This shows that considering the connectivity between voxels can improve blood vessel segmentation results

**Table 4: Ablation study on our architectures. √ indicates the presence of this block, with the best results highlighted**

| Basnet | DE | DI | DSC | HD95 | ASSD | REV | SENS |
|---|---|---|---|---|---|---|---|
| √ | | | 90.469 | 3.577 | 0.954 | 8.699 | 87.098 |
| √ | √ | | 91.449 | 1.433 | 0.750 | 3.662 | **92.739** |
| √ | √ | √ | **92.413** | **1.188** | **0.669** | **2.581** | 91.331 |

*4.5.2 Ablation study on architectures.* To demonstrate the effectiveness of each block in TransConNet, we present quantitative comparison results against other relevant architectures. We start with the Basnet and progressively enhance it with various blocks, including the direction excitation(DE) block and direction interactive(DI) block. Table 4 showcases the outcomes of this ablation study.Basnet only contains Bi-level Former block and skip connection

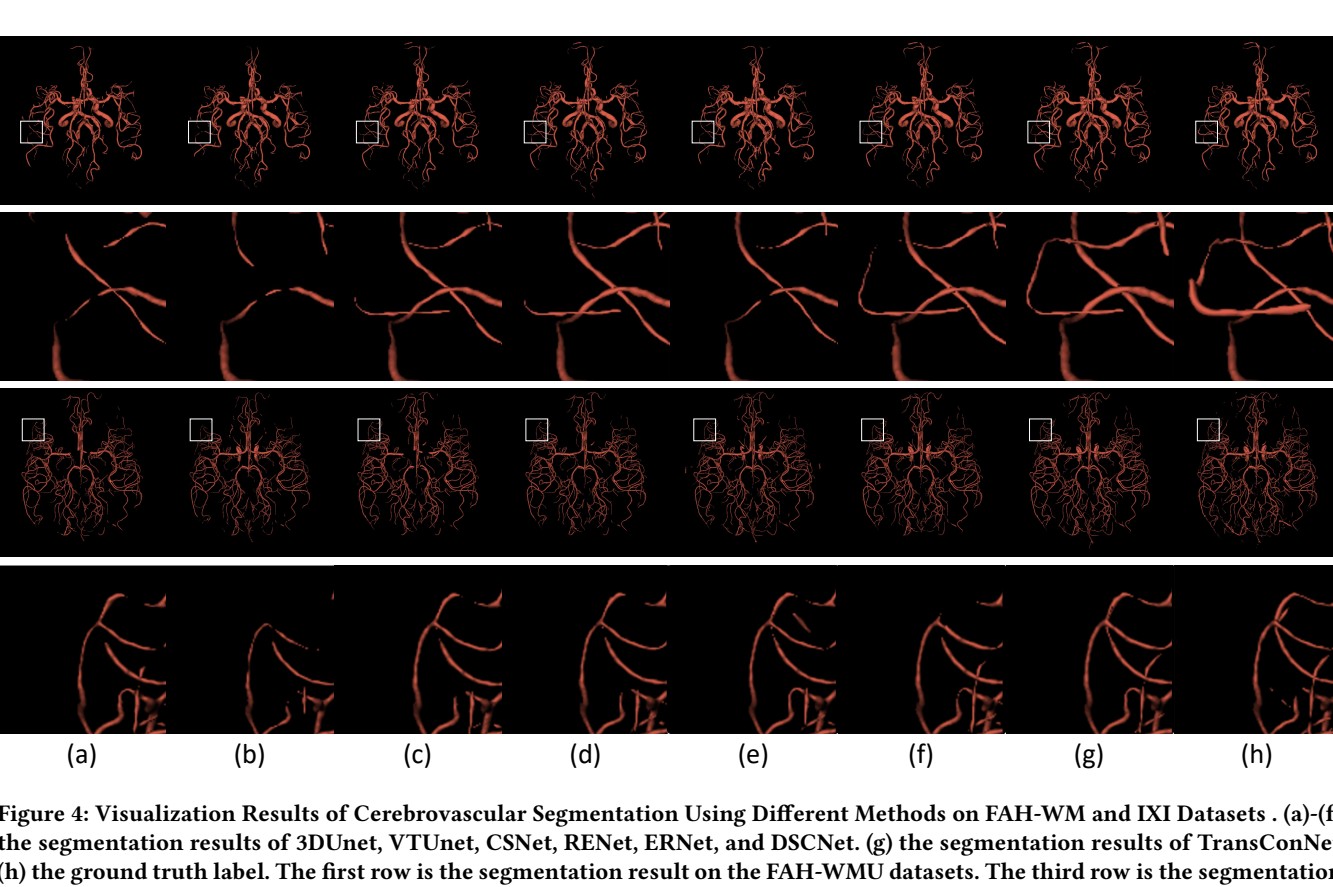

**Figure 4: Visualization Results of Cerebrovascular Segmentation Using Different Methods on FAH-WM and IXI Datasets . (a)-(f) the segmentation results of 3DUnet, VTUnet, CSNet, RENet, ERNet, and DSCNet. (g) the segmentation results of TransConNet. (h) the ground truth label. The first row is the segmentation result on the FAH-WMU datasets. The third row is the segmentation result of the IXI datasets. The second and fourth rows are partially enlarged images, respectively.**

In the second row of Table 4, the absence of the direction excitation block notably affects the connectivity loss. To ensure a fair comparison, we introduce an additional upsampling output at the bottleneck. It becomes evident that integrating the direction excitation block yields improvements in segmentation results, with the DSC increasing by 0.98%, the HD95 decreasing by 2.414 mm, and the ASSD decreasing by 0.204 mm, respectively. Exciting directional information from feature maps facilitates the network's ability to learn the directional information inherent in the connectivity mask.

Furthermore, the integration of direction interactive block results in notable enhancements, with DSC showing a boost of 0.964%. Concurrently, HD95 observes a reduction of 0.245 mm, and ASSD experiences a decrease of 0.081mm. The incorporation of a direction interactive block injects directional information into the skip connections, simultaneously ensuring the establishment of vessel connectivity while enhancing the resolution of the feature map.

*4.5.3 Ablations study on training loss.* We concurrently augment each loss in $L_{main}$ and $L_{aux}$. Initially, we utilize $L_{BCE}$ and $L_{DICE}$ as the foundational loss functions. Upon adding $L_{Iou}$, we observed a 1.216% increase in DSC. However, the disparities between HD95 and ASSD were not significant. This suggests that additional loss functions enhance segmentation efficacy, yet challenges persist in accurately delineating vessel edges.

By adding the connectivity loss $\mathcal{L}_{BCE}(C, G_C)$ i.e. $\mathcal{L}^1_{BCE}$, we observed a marginal increase of 0.029% in DSC, a decrease of 0.145mm

**Table 5: Ablation study on training loss. √ indicates the presence of this loss, with the best results highlighted**

| $\mathcal{L}_{main} + \mathcal{L}_{aux}$ | | | | | Metrics | | |
|---|---|---|---|---|---|---|---|
| $\mathcal{L}_{Dice}$ | $\mathcal{L}_{BCE}$ | $\mathcal{L}_{Iou}$ | $\mathcal{L}^1_{BCE}$ | $\mathcal{L}^2_{BCE}$ | DSC | HD95 | ASSD |
| √ | √ | | | | 90.978 | 1 .399 | 0.761 |
| √ | √ | √ | | | 92.239 | 1.401 | 0.740 |
| √ | √ | √ | √ | | 92.268 | 1.256 | 0.695 |
| √ | √ | √ | √ | √ | **92.413** | **1.188** | **0.669** |

in HD95, and a reduction in ASSD. While the overall improvement in segmentation may not be particularly pronounced, including connectivity loss contributes to a more precise delineation of blood vessel boundaries by mitigating the negative impact of IoU loss. Upon adding $\mathcal{L}_{BCE}(\tilde{C}, G_C)$ i.e. $\mathcal{L}^2_{BCE}$, all segmentation metrics are further optimized.

Upon the above analysis, we observed that including connectivity loss enhances the network's ability to segment blood vessel boundaries, resulting in more precise delineations. This improvement is valuable for detecting diseases occurring at the cerebrovascular boundary.

# 5 CONCLUSION

In this study, we introduced TransConNet, a connectivity-based cerebrovascular segmentation network. By converting traditional segmentation masks into connectivity masks, we transformed the voxel classification task into one based on connectivity prediction, thereby enhancing the preservation of cerebrovascular connectivity. The building of 3D Bi-level Former blocks improved computational efficiency in sparse vessel feature extraction, while the integration of 3D-direction excitation and interactive blocks enhanced the capture and utilization of directional information.Our experimental results demonstrate the superiority of TransConNet in preserving cerebrovascular connectivity compared to existing methods. This has significant implications for preoperative examinations in computer-assisted medical procedures, where accurate segmentation and preservation of vessel connectivity are paramount. In the future, we'll refine directional excitation and interactive blocks to boost vessel segmentation accuracy and connectivity even further. This advancement will likely entail refining algorithms,and possibly integrating cutting-edge technologies like machine learning or advanced image processing techniques. The goal is to achieve more precise segmentation, which is crucial for various medical imaging applications such as diagnosis and treatment planning.

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
