# OpenReview forum: "Connectivity-based Cerebrovascular Segmentation in Time-of-Flight Magnetic Resonance Angiography"
_acmmm.org/ACMMM/2024/Conference — MM2024 Poster_

### Official Review · Reviewer_b5Lb · 2024-05-24

**Rating:** 3
**Confidence:** 4

**Summary:**

This article proposes a TOF image segmentation method based on voxel connectivity, introducing directional excitation and interactive modules to enhance the network's performance. The model's effectiveness has been validated on multiple datasets.

**Strengths:**

The topology and connectivity of blood vessels are important features. Constructing a new optimization objective based on connectivity is an excellent innovation. The multiple improvement modules proposed on this basis enrich the content of the work presented in the article.

**Limitations:**

- In line 224, "divides the input images into non-overlapping patches of size 4×4×4" seems to be incorrect.

- In line 272, W×H×S×C, S has never been mentioned before.

- In Figure 3, the meaning depicted by the author is completely inconsistent with Equation 15, which can be misleading. I suggest revising it.

- In line 473, there is a lack of definition for b in "first b largest values." The author should provide the value of b.

- The combination of backbone + DI should be added to the ablation study in Table 4 to verify the effectiveness of the DI module on network performance.

- I believe it is necessary to ablate Loss(aux) to verify the effectiveness of the auxiliary loss.

- Adding one or two models that also use connect-based segmentation in the comparative experiment would be more convincing.

- I believe this work has considerable innovation and sufficient workload; however, the poor illustrations in Figures 1-3, along with numerous writing errors in the paper, have significantly weakened its quality. I strongly recommend that the authors reorganize the article's structure and redraw the figures.

**Suitability:**

2

---

### Official Review · Reviewer_pdRt · 2024-05-24

**Rating:** 4
**Confidence:** 3

**Summary:**

This manuscript proposed a connectivity-based cerebrovascular segmentation framework, TransConNet, that addresses the main challenges of cerebrovascular segmentation by incorporating a connectivity mask.

**Strengths:**

1.	It proposes a new network model TransConNet to solve the cerebrovascular segmentation problem. This network is an innovation of the ordinary Transformer model, adding sparse attention mechanism, 3D directional excitation block, 3D directional interaction block and transform the task of voxel classification into the task of predicting inter-voxel connectivity.
2.	This scheme can solve the discontinuity problem of fine blood vessel position, and improve the computational efficiency of feature extraction and the capture and utilization of directional information, so as to further improve the accuracy and connectivity of blood vessel segmentation.
3.	In the experimental part of the study, multiple evaluation indexes were set up to fully evaluate the model, and the spatial overlap and geometric accuracy were fully considered.
4.	The proposed scheme outperforms the state of the arts and has an important application prospect in the preoperative examination of computer aided medical procedures.

**Limitations:**

1.	In the abstract, it is suggested to provide detailed data support for the description of the results.
2.	The title of the method section incorrectly wrote "METHODS" as "METHDOS". Please check the manuscript carefully.
3.	At the end of the introduction section, it is suggested to describe the general structure of the article.
4.	Picture description not clear. In the description of Figure 2 (a), there is a lack of explanation of the specific meaning of parameters W, H and S and their corresponding values.
5.	In Eq.18, It is recommended to introduce the meaning of the coefficient 0.3 before Laux.
6.	In the part of experiment details, the assignment of training dataset and validation dataset is introduced, and the test dataset is not mentioned. However, the subsequent experimental results are based on the average value of the test dataset. Is the verification and the test used the same part of dataset?It is recommended to clarify this issue. Now that the validation dataset is set, what validation method is used? In addition, this section does not describe the setting of hyperparameters in detail, and does not explain the setting of epoch and batch size.
7.	It is recommended not to compare with unpublished papers, such as those on ArXiv. In the comparative experiment, the source of the classical segmentation method VTUet[17] is a paper on arXiv.
8.	In the ablation study, it was not clear which dataset was used here. In addition, the analysis of experimental results is incomplete. In ablations study on connectivity mask and connectivity loss, ASSD values increased slightly after con mask was added. What are the possible reasons? Was it the result of chance? Why are the results obtained when seg mask and con mask are added at the same time different from the results shown in the previous two data sets? In addition, the reason why the SENS index results of Basnet+DE were significantly better than those of the other two cases in the ablation study on architectures was not analyzed.
9.	In the ablation study on architectures, mention“To ensure a fair comparison, we introduce an additional upsampling output at the bottleneck……”It is suggested to clarify what exactly is meant by possible unfairness here. Why does add upsampling output work?
10.	From an overall perspective of the article, it is suggested that the author includes  the exploration and thinking about the limitations of the proposed research method at the conclusion.

**Suitability:**

2

---

### Official Review · Reviewer_Lr2R · 2024-05-24

**Rating:** 3
**Confidence:** 1

**Summary:**

The article proposes a connectivity-based method for brain vessel segmentation, converting voxel classification into predicting voxel connectivity by modeling connectivity. Sparse 3D dual-layer routing attention is employed to reduce computational costs while effectively capturing brain vessel features. Additionally, 3D directional excitation blocks are designed to enhance directional information in the feature maps.

**Strengths:**

The description of the theory in the article is very specific and professional, demonstrating a decent level of scientific research. The article has a clear logic and is written in a standard format.

**Limitations:**

1.In the Related Work section of the article, the author did not introduce much research on the use of transformer models for cerebral vascular segmentation. Here, some relevant content can be added.
2.In the introduction of the Method in the article, the first part is about modeling connectivity, lacking some basic content. It is suggested that the author add some detailed descriptions here.
3.In section 4.4. The author has conducted a large number of comparative experiments, but lacks sufficient analysis and explanation. The author should provide a detailed interpretation of the experimental results based on the structure of different comparative networks.
4.The author presents fewer intuitive views in the article. It is recommended that the author add visual views of different segmentation results corresponding to the ablation experiment to expand the visualization effect of the article.

**Suitability:**

2

---

### Official Review · Reviewer_uFPs · 2024-05-24

**Rating:** 4
**Confidence:** 2

**Summary:**

The primary contribution of this article is the proposal of a connectivity-based 3D segmentation method for small cerebral blood vessels in MRI images, aiming to prevent the occurrence of fragmented vessels. It tranfers the task of cerebrovascular segmentation from voxel classification to predicting inter-voxel connectivity, guaranting the integrity of small blood vessels. A lot of Experiments verify its claims.

**Strengths:**

1. The paper tranfers the task of cerebrovascular segmentation from voxel classification, guaranting the integrity of small blood vessels. The novelty is somewhat limited but I feel that it is enough to justify publication.

2. A lot of experiments were conducted to verify its contribution. In particular, ablation experiments confirmed the improvement effect of each module.

3. The paper is well organized and clear.

**Limitations:**

1. I am worried that the scope of this paper is not related to multi-media. It is based on the unimodal of 3D MRI.

2. The paper uses a lot of various metrics to validate the segmentation results of cerebral blood vessels, including Dice, HD, etc. But, the paper focuses on the connectivity of the small blood vessel. It suggested to contain connectivity-related metrics to evaluate segmentation results, like NoS metric in [1].

3. More classic segmentation methods should be added for comparison, like nnUnet [2].

[1] @misc{yang2023segmentation,
    title={Segmentation and Vascular Vectorization for Coronary Artery by Geometry-based Cascaded Neural Network},
    author={Xiaoyu Yang and Lijian Xu and Simon Yu and Qing Xia and Hongsheng Li and Shaoting Zhang},
    year={2023}
}

[2] @article{isenseeNnUNetSelfconfiguringMethod2021,
  title = {{{nnU-Net}}: A Self-Configuring Method for Deep Learning-Based Biomedical Image Segmentation},
  shorttitle = {{{nnU-Net}}},
  author = {Isensee, Fabian and Jaeger, Paul F. and Kohl, Simon A. A. and Petersen, Jens and Maier-Hein, Klaus H.},
  date = {2021-02},
  journaltitle = {Nature Methods},
  shortjournal = {Nat Methods},
  volume = {18},
  number = {2},
  pages = {203--211},
  issn = {1548-7091, 1548-7105},
  doi = {10.1038/s41592-020-01008-z},
  url = {http://www.nature.com/articles/s41592-020-01008-z},
  urldate = {2021-07-23},
  langid = {english}
}

**Suitability:**

1

---

### Meta-Review · Area_Chair_rmTv · 2024-06-30

**Recommendation:** Accept (Poster)
**Confidence:** 5

**Metareview:**

This paper proposed a connectivity-based cerebrovascular segmentation framework by incorporating a connectivity mask. After rebuttal, four borderline accepts are recommended. Given the consensus of reviewers, a decision of accept is suggested. I do suggest the authors revise the manuscript by taking all the reviewers’ comments into consideration, especially on the experiments and further illustration on details.